



# Defining scale thresholds for geomagnetic storms through statistics

Judith Palacios[1], Antonio Guerrero[1], Consuelo Cid[1], Elena Saiz[1], and Yolanda Cerrato[1]

[1]Space Weather Group, Departamento de Física y Matemáticas, Universidad de Alcalá, Alcalá de Henares, Spain.

*Correspondence to:* Judith Palacios: judith.palacios@uah.es

**Abstract.**

Geomagnetic storms, as part of the Sun-Earth relations, are continuously monitored with different indices and scales. These indices have some scale thresholds to quantify the severity or risk of geomagnetic disturbances. However, the most usual scale thresholds are arbitrarily chosen. In this work we aim to quantify the range of the thresholds through a new method. These new
thresholds are based on statistical distribution fitting.

The data used are from a regional real-time high-cadence geomagnetic index, named *LDiñ*, and its derivative, *LCiñ*. We considered the negative part of *LDiñ*, as significant for geomagnetic disturbances; and the absolute value of *LCiñ*, significant for geomagnetically induced currents. Then we look for the best fit for different statistical continuous distributions applied to these indices.

The method yields that the beta prime is the most suitable functions for negative values of *LDiñ*, whereas power-law and Johnson-SU are the best fits for *LCiñ* and the whole distribution, respectively. We define new thresholds for intense, very intense and extreme geomagnetic disturbances as the intersects between these best fit distributions and the index complementary cumulative distribution function.

Then, thresholds for the negative part of *LDiñ*, are -100 nT, -205 and -475 nT. The thresholds for the absolute value of *LCiñ*,
are 6, 18 and 32 nT min$^{-1}$. The thresholds defined here provide criteria to assess the vulnerability to geomagnetic activity on design or mitigation purposes.

These threshold definitions will be applied for different products in the Spanish Space Weather Service (SeNMEs) website http://www.senmes.es/index-en.php.

## 1 Introduction

'Solar storms', as an example of natural hazards, is a rough term which comprises unspecified cases of eruptive and energetic solar events, such as coronal mass ejections (CMEs), flares, and also events of accelerated particles. The Sun as an active star due to its magnetic field, do generate these eruptive instances – along with the solar wind–, and when CMEs are Earth directed, impinges our planet's shield, that is the Earth's magnetic field. The perturbation on the Earth's environment, named generally geomagnetic storm (or more specifically, geomagnetic disturbance) can be observed as a decrease of the Earth's magnetic field.
They are typically measured with magnetometers in geomagnetic observatories spread across the globe.



Quantifying geomagnetic storm intensities is one of the main aims in space weather. Extreme space weather events are treated now as natural environmental hazards since they pose a serious threat to different technologies, including critical infrastructures. The threats include power grid upsets, that can be damaged due to sudden changes in the Earth's magnetic field, originating geomagnetically induced currents (GICs); problems in GNSS positioning; orbit displacement of satellites, or

radio blackouts, among other undesirable effects. A summary of these risks and their economic and societal impact appeared recently in Eastwood et al. (2017). Very recently, macro-economic losses have been modelled considering worst-case scenarios centered in different areas of the globe (Schulte in den Bäumen et al., 2014).

Different geomagnetic indices have been created to account for different geophysical phenomena, and then, a scale was defined accordingly. Scales may rely on the physics behind an index, or a scale may be effect-based only. Some examples of

the kind of scales are for earthquakes – effect-based Mercalli scale, and the $M_w$ scale, that is physics-based; for nuclear risks, the INES scale is an effect-based, and the NAMS scale is more physical (Wheatley et al., 2017). Below, we comment the most commonly employed indices for space weather.

The *Dst* is a geomagnetic index, considered to provide a quantitative measurement of geomagnetic disturbances at low latitudes. The method for the elaboration of the *Dst* index is described in http://wdc.kugi.kyoto-u.ac.jp/dstdir/dst2/onDstindex.

html and it is assumed as a representation of the ring current enhancement encircling the Earth. This index is widely used in space weather. However, the *Dst* index has been questioned to misrepresent the ring current, not accounting for asymmetries (Grafe, 1998); or being inaccurate due to important seasonal offsets (Karinen and Mursula, 2006), that may even lead to a mismatch in the currently used scale.

In Gonzalez et al. (1994), the intensity of a storm was established by the minimum *Dst* value, and the scale thresholds of

moderate and intense storm were also set. The definition for moderate storm comes from a threshold placed at $Dst$ percentile 8; and the intense storm definition comes from the $Dst$ percentile 1, employing a dataset of 10 years for these thresholds. The superintense threshold consideration as the value $Dst <= $ -250 nT (Tsurutani et al., 1992, and references therein) and employed after by e.g., Echer et al. (2008a, b) comes from a case-study ad-hoc threshold. These values are tabulated in Table 1 among other indices thresholds, listed below.

The *K* (3-hour) and *A* (24-hour) indices (Bartels et al., 1939) have been used for decades to convey the intensity of major geomagnetic disturbances. The *K* index ranges for 0 to 9 and it is based on the maximum range in magnetic field variation over 3-hour interval, in a quasi-logarithmic scale. The *A* index is a 24-hour index derived from the average of the eight 3-hour indices during the day. The planetary $K_p$ index is an average of the *K* index from selected ground-based magnetometers with global coverage (Bartels and Veldkamp, 1949). Similarly, its daily (24-hour) version is called $A_p$. A number of limitations of

these indices is described in Rostoker (1972), such as the arbitrary upper limit of *A*, or the diurnal variation that exhibits a seasonal component for *K*.

In 1999, the U.S. National Oceanic and Atmospheric Administration (NOAA) issued a five-level geomagnetic storm scale named *G* scale (http://www.swpc.noaa.gov/noaa-scales-explanation) that classifies geomagnetic disturbances with an effect-based scale intended on physical infrastructures, mainly for power systems. This effect-based scale stems from one of the two

indices already used to measure the severity of these disturbances, that is the $K_p$ (Bartels and Veldkamp, 1949).



**Table 1.** This table shows the thresholds of geomagnetic activity for different indices

| Index | Quiet-minor | Moderate storm | Intense storm | Superintense |
|-------|-------------|----------------|---------------|--------------|
| $K_p$ | 0-4 | 5 | 7 - 9 | |
| $A_p$ [2nT] | 0 - 20 | 30 - 50 | 100 - 400 | |
| $Dst$ [nT] | >−50 | −50 - −100 | <−100 | <−250 |

For more information about these geomagnetic indices, one can refer to Bartels et al. (1939); Rostoker (1972); Mayaud (1980).

All this variety may pose an additional complication for the user to differentiate between physical and effect-based scales, their thresholds and their applicability range. In the end, proper threshold definition is necessary for considerations on the importance of geomagnetic disturbances and for further utilization by users and decision makers.

Some of these global geomagnetic indices are not fine tuned for the current space weather requirements, such as high temporal cadence. In addition to that, caution is recommended due to the unevenly spanned observatories around the globe (Rostoker, 1972; Karinen and Mursula, 2006). Other of the main drawbacks of most of these geomagnetic indices is that they are averages of measurements from a network of different geomagnetic observatories. This fact may provoke averaging out of important local geomagnetic disturbances and therefore, fundamental information may be missing. Rather than the consideration of a geomagnetic storm quantification as global, it is proved that very local effects may happen, such as H-spikes or Carrington-like disturbances (Cid et al., 2015; Saiz et al., 2016). The consequences of local disturbances are of the maximum relevance, since these large peaks are linked to large time derivative values, which are related to GICs.

A new geomagnetic index, $LDi$, has been developed to overcome the problems of lack of temporal resolution, and the risk of averaging out spikes due to the use of several observatory data. This real-time high cadence local geomagnetic index has been designed to monitor the geomagnetic disturbances at any mid-latitude location. This index is created to remove the solar regular daily geomagnetic variation and other trends. The importance of local and high temporal resolution indices is substantial. The $LDi$ obtained from Spanish geomagnetic records is named $LDi\tilde{n}$ (Guerrero et al., 2016).

Furthermore, the derivative of $LDi\tilde{n}$, named $LCi\tilde{n}$, is highly correlated with the geomagnetically induced currents records from the Spanish Power Company (Cid et al., 2016). Being aware of these facts, and knowing that accurate real-time monitoring according to the needs of the final users is key, the Spanish Space Weather Service (SeNMEs, http://www.senmes.es/index-en.php) in 2014 introduced the G- and C-scale, for $LDi\tilde{n}$ and $LCi\tilde{n}$, respectively.

Thresholds are not only employed to classify the severity of a natural hazard but also employed for computing the waiting time and time occurrence of events. The thresholds used for several types of natural hazards such as earthquakes, volcanic eruptions or floods are based on 1 in 50- or 1 in 100-year extremes; this approach for space weather is also applicable, as shown in different works, e.g., Love (2012); Nikitina et al. (2016); Love et al. (2016); Wintoft et al. (2016), among others.

In this paper, we use different distribution functions seeking the best fit for the regional index $LDi\tilde{n}$ and its derivative $LCi\tilde{n}$; then, we define thresholds for this index and its derivative. Fitting distributions aiming at recurrence or waiting times





estimations is out of the scope of this work. Opposite to the vast majority of the mentioned references, we do not use thresholds to help fitting a part or the whole distribution; threshold rationale and distribution properties are the main purposes. Hence, one of the key goals in this paper is aiming at overcoming some of the arbitrariness to establish thresholds for geomagnetic disturbances and its rate of change through the objective method explained in this paper. Its application can be extended to any

other global or regional index.

The paper is structured as follows: data used are presented in Section §2, along with the preliminary statistical considerations; next in Section §3, the method is described, obtaining the best fit distributions to the indices in Section §3.1; and then in Section §3.2, the thresholds for these indices are defined by these statistical best distribution intersects. The results and discussion are presented in Section §4, and Conclusions appear in Section §5.

## 10   2   Geomagnetic data

We used *LDiñ* data computed from SPT geomagnetic observatory (San Pablo de los Montes, Toledo, Spain) from 1997 to 2012 – solar cycles 23 and 24, comprising two rising phases and one solar maximum –, processed in such a way to remove the daily variation in the $H$ component of the geomagnetic field, which is not straightforward for mid-latitude data. The processing method, which is now patent pending (Guerrero et al., 2016), will be actually detailed in a forthcoming paper.

Therefore we consider *LDiñ* as this local geomagnetic index after this specific processing, whose units are nT and its temporal cadence is 1 minute. We define hereafter *LCiñ* as the temporal derivative of *LDiñ*, in nT min$^{-1}$. Since the goal of this paper is not the storm recurrence or waiting time, not timing between peak values is considered. The whole dataset are 1-min values without any further consideration of recurrence.

We computed more than 15 years of data. This period is chosen since it is almost the whole period in which SPT has their

data digitized. Also, it compresses the solar cycle 23 and its very intense geomagnetic storms. The *LDiñ* and its temporal derivative along almost 16 years are plotted in Fig. 1. The values are upper limited (in absolute value) to the highest peak values of the geomagnetic disturbances during the solar cycle 23. Values of the derivative are also constrained by these storms.

To give a hint about the value distributions of *LDiñ* and *LCiñ*, we plotted the histograms, normalized to the total amount, in Fig. 2. The bin size is 5 nT for *LDiñ* and 0.5 nT min$^{-1}$ bin size for *LCiñ*. On geomagnetic storms, negative values actually

define the storm since they mean a horizontal component depression of the geomagnetic field due to a ring current enhancement; therefore we will take into account only these negative values, naming it as $|neg(LDi\tilde{n})|$. It is evident that this distribution is not gaussian but other kind of distribution, with a long tail and very skewed to negative values, as shown in in Fig. 2 (top left).

We consider that the positive and negative values of *LCiñ* are meaningful because both may be related to the geomagnetically induced current (GIC) generation, interesting for space weather purposes. Therefore, we will consider this possibility equally

for both signs by studying the absolute value of *LCiñ*, naming it $|LCi\tilde{n}|$. As shown in Fig. 2 (top right), the distribution of *LCiñ* has a very small skewness.

Taking into account the significant values in the form of $|neg(LDi\tilde{n})|$ and $|LCi\tilde{n}|$, we plotted *log-log* histograms. In the case of $|neg(LDi\tilde{n})|$, the heavy tail of the distribution is shown clearly in the bottom left of Fig. 2. A conspicuous linear trend





appears for $|LCi\tilde{n}|$ histogram in the bottom right panel of Fig. 2. We note the bin size variation in logarithmic scale, issue that will be improved in the next Section.

We also show a scatterplot of the absolute value of *LDiñ* and *LCiñ* to see how their values relate to each other and show the relationship, if any, between extreme values. Unfortunately we were not able to fit a line of this scatterplot due to the large

amount of points, similar to those in Tóth et al. (2014). Due to the large dataset, we created a two-dimensional histogram instead, plotting the absolute values of *LDiñ* and *LCiñ*, in the left panel of Fig. 3. The bin size is equivalent to $\approx 6$ nT for *LDiñ* and $0.5$ nT min$^{-1}$ for *LCiñ*. The histogram evidences that large values of $|LCi\tilde{n}|$ may happen during non-highly disturbed geomagnetic conditions, according to *LDiñ* values. An equivalent plot is made in $log - log$ in the right panel of Fig. 3, with the same bin amount that the previous one. The truncation in the right plot is at $0.1$ nT and $0.1$ nT min$^{-1}$. Datapoints cluster

roughly around 0 for both indices (see red area), which is the noisy data regime. Interestingly, values also clutter in regions (shown in green) where some trends between *LDiñ* and *LCiñ* can be noticed. The areas displayed in blue are the most extreme values and they are more scattered.

## 3  Distribution fitting and threshold definition

### 3.1  Statistical distribution fitting

A statistical approach is used in this Section. To find the most appropriate distribution for these meaningful defined indices as $|neg(LDi\tilde{n})|$, $|LCi\tilde{n}|$ and the whole *LDiñ* distribution, we use the Python stat package, looking for the most appropriate distribution fit. We tested 84 continuous functions implemented in Python (the whole list includes 94: https://docs.scipy.org/doc/scipy-0.18.1/reference/stats.html) plus a powerlaw distribution function package described later. Most of these functions can be found, e.g, in Wilks (2011). The not used distributions due to convergence lack are the following: *burr12*, *exponnorm*,

*gennorm*, *halfgennorm*, *invnorm*, *kappa4*, *kappa3*, *levy_stable*, *skewnorm*, *trapz*, *vonmises_line*.

These fits and the subsequent intersection points between significant distributions are utilized to define the threshold for intense, very intense and extreme cases. Note that no previous threshold to avoid the distribution bulky part is utilized to fit the functions. All the daily variation usually appears as noise in a histogram. Sometimes the distribution bulk is not regarded for fitting distribution functions; here, since the noise is highly reduced due to the *LDiñ* processing, the distribution bulk is also

suitable for fitting, not only the distribution tail.

The minimization has been done through the Kolmogorov-Smirnoff (KS) distance parameter *D* computation. The minimization is parametric, through the maximum likelihood estimator (MLE) used to perform the distribution fitting as default; as a double check, we also inspected the minimization by the Kolmogorov-Smirnoff method, producing the same results. Consequently, the statistics results are robust. Moreover, the fitting is not binning dependent.

Looking for the best fit, we seek the minimum value of the Kolmogorov-Smirnoff distance *D* for all the distributions for $|neg(LDi\tilde{n})|$, $|LCi\tilde{n}|$ and the whole *LDiñ*. Trying to improve the histogram plots, we used the complementary cumulative distribution function (CCDF). Distribution function plotting is made by the survival function computation. The CCDF is appropriate to be fit since it avoids bin dispersion, obtaining an appropriate binning at the distribution tail (as in Riley, 2012).



This bin dispersion appears, for instance, in Fig. 2 at the bottom plots. For now on, the bin size employed from $|neg(LDi\tilde{n})|$ is 5 nT and for $|LCi\tilde{n}|$ is 2 nT min$^{-1}$.

For the $|neg(LDi\tilde{n})|$ computed CCDF, the best fit is achieved when $D$ is in the order of magnitude of $10^{-3}$. These distributions are beta-prime (beta of the second kind), the $f$ or Fisher-Snedecor (a type of beta-prime), and the Mielke (beta-
kappa) distribution. Next, $D$ values are around 0.01 for the Burr distribution (a generalized log-logistic type), exponenciated Weibull ('exponweibull'), generalized Gamma distribution ('gengamma'), powerlaw lognormal ('powerlognorm') and generalized Pareto distribution ('genpareto') and Lomax (a Pareto type-II distribution). Upper $D$ values around 0.02 correspond to the lognormal distribution, and to three other distributions (not shown, and this is marked as [...]). These values are shown in Table 2. The best ten fit distributions are displayed in Fig. 4 *(top)*.

For the case of $|LCi\tilde{n}|$ computed CCDF, the best fit distribution is the power law, obtained with the Python package 'powerlaw' (Alstott et al., 2014). The starting point of the powerlaw $x_{min}$ is computed by minimization of the $D$ parameter: $D$ is minimum at $x_{min}$=1.55 nT min$^{-1}$. Then, the powerlaw function parameters are computed. No upper value ($x_{max}$) is considered for the fit. The powerlaw index $\alpha$ is equal to $\alpha$=3.339±0.007, noting the negative slope. Note, as warned in Alstott et al. (2014) that $|LCi\tilde{n}|$ CCDF has a slope of $-(\alpha$-1) and renormalized. That is the reason why is called 'shifted CCDF' in the
Fig. 4 *(middle)*. The powerlaw use and its low $D$ confirms the first impression on the linear trend of the bottom right panel in Fig. 2.

In addition to the power law, the Mielke (beta-kappa) is the best fit. Next, the rest of the distributions exhibit a $D$ larger than 0.02, as the Gilbrat (from the lognormal family), Fisk (log-logistic), and the lognormal distribution. The rest of functions appear in Table 3. Unfortunately we did not find any distribution that fit the $|LCi\tilde{n}|$ with $D$ in the order of magnitude of $10^{-3}$
apart from powerlaw. The best ten fit distributions are displayed in Fig. 4 *(middle)*.

For the whole *LDiñ* CCDF case, it is more complicated to fit both positive and negative parts. However, the Johnson-Su (Johnson's unbounded, a hyperbolic sine type) is the best distribution, with $D$ of the order of magnitude of $10^{-3}$. Also the non-central Student's $t$-distribution ($nct$) is in the same order. Values of $D$ larger than 0.03 correspond to the Student's $t$-distribution, the generalized logistic distribution ('genlogistic'), the hyperbolic secant ('hypsecant'), the Laplace, the double
Gamma distribution ('dgamma'), double Weibull distribution ('dweibull'), and the logistic. The Cauchy distribution corresponding $D$ is larger than 0.06, and we can see that is departing from the best fitting, as shown in Table 4. The best ten fit distributions are displayed in Fig. 4 *(bottom)*.

For completeness, we mention here the best fit for the whole *LCiñ* distribution, which is the Johnson Su distribution, with $D$=0.007; the Student's $t$ has $D$=0.011; and also the non-central Student's $nct$ gets similar values, $D$=0.011– confirming the
*LCiñ* very small skewness –; the next best fit is the double Weibull, with $D$=0.026, and Cauchy distribution with $D$=0.028; therefore general trends are very similar to the whole *LDiñ* distribution.



**Table 2.** $D$ values for best fitting of distributions for the $|neg(LDi\tilde{n})|$.

| KS value $D$ | corresponding distribution |
|:---:|:---:|
| 0.006 | betaprime |
| 0.006 | $f$ |
| 0.009 | Mielke |
| 0.010 | Burr |
| 0.012 | exponweibull |
| 0.013 | gengamma |
| 0.013 | powerlognorm |
| 0.017 | genpareto |
| 0.017 | Lomax |
| ... | ... |
| 0.023 | lognorm |

**Table 3.** $D$ values for best fitting distributions for $|LCi\tilde{n}|$.

| KS value $D$ | corresponding distribution |
|:---:|:---:|
| 0.004 | powerlaw |
| 0.018 | Mielke |
| 0.022 | Gilbrat |
| 0.024 | Fisk |
| 0.027 | lognorm |
| 0.027 | Johnson-Su |
| 0.028 | invgamma |
| 0.029 | Betaprime |
| 0.029 | invgauss |
| 0.030 | powerlognorm |

## 3.2 Defining thresholds through distribution best fits

As a first approach, we established some arbitrary thresholds in a preliminary way. For $|neg(LDi\tilde{n})|$, these thresholds we used are the percentiles 99%, 99.9% and 99.99%. The corresponding values to these percentiles are -87, -179 and -357 nT, for intense, very intense and extreme, respectively.

5    In the case of $|LCi\tilde{n}|$, setting the thresholds to percentiles 99.9%, 99.99% and 99.999%, the corresponding values are 5, 13, 30 nT min$^{-1}$. Percentiles are employed, for instance, for the mean residual life of the $H$ derivative for a high latitude observatory, as in e.g., Thomson et al. (2011).



**Table 4.** $D$ values for best fitting distributions for the whole $LDi\tilde{n}$.

| KS value $D$ | corresponding distribution |
| --- | --- |
| 0.006 | Johnson-Su |
| 0.007 | non-central $t$-distrib |
| 0.031 | $t$-distrib |
| 0.037 | genlogistic |
| 0.041 | hypsecant |
| 0.045 | laplace |
| 0.045 | dgamma |
| 0.045 | dweibull |
| 0.046 | logistic |
| 0.067 | Cauchy |

However, an advanced way of defining thresholds is making use of the statistical distributions and treatment presented in Section §3.1. For all distributions, we can see that some parts of them have excellent fits and some others whose fitting is not that good, particularly in the distribution tails. Then, taking advantage of this fact, a way of defining the threshold is the intersects of the CCDF with the best fit distribution, or even the intersect of the two best fit distributions. Then, we can define

a threshold of a determined degree of geomagnetic disturbance – or its derivative – when the intersect is in the bulk, the tail or the extreme tail.

For the $|neg(LDi\tilde{n})|$, as shown in Fig. 4 *(top)*, the intersects are computed with a relative tolerance of 0.005. Both betaprime (black dashed line) and $f$ (thick red line) intersects the CCDF (in blue) in -100 nT, -205 nT and -475 nT. These values can be set as the thresholds for intense, very intense and extreme disturbances. Lognormal (dashed yellow line) starts departing in

-30 nT, and most of the others start departing between -60 and -75 nT, fitting well the bulk but not meeting the heavy tail. This regime difference may mark the threshold setting for moderate disturbances at about -60 nT.

For the case of $|LCi\tilde{n}|$, – Fig. 4 *(middle)* – there are two intersects of $|LCi\tilde{n}|$ shifted CCDF (green line) and the fit powerlaw (in red), at about 6 nT min$^{-1}$, which is where powerlaw departs from the shifted CCDF ; and at 18 nT min$^{-1}$, where the powerlaw red line meets again the shifted CCDF. We can set the two thresholds in the powerlaw for the case of unsettled

and intense values in these two intersects. Next, we can consider the intersect between some the best $D$-scoring distributions, as Gilbrat (brown line) and betaprime (dashed green line) to determine the threshold for the extreme events. In this case, the intersect is found at 32 nT min$^{-1}$, with an absolute tolerance of $10^{-10}$. The lognormal departs the CCDF at about 6 nT min$^{-1}$, clearly separating the bulk from the tail.

The whole $LDi\tilde{n}$ – Fig. 4 *(bottom)* – is fit for the sake of illustrative purposes on asymmetric distributions, but the threshold

in this case is not sought out.





## 4   Discussion

An amount of research about distributions applied on different data has been done, such as *Dst* mainly but also on SYM-H, polar indices, etc. Historically, one of the first applied distributions which may fit the very skewed distribution of geomagnetic disturbances is the lognormal.

Lognormal distribution shapes of the *Dst* have been commented in Campbell (1996), somehow relating the *Dst* shape profile (that resembles a lognormal function) to a lognormal distribution. In Aguado et al. (2010), the recovery phase is fit as a hyperbolic function. However, the disturbance profile shapes will not be further discussed here.

Importantly, also quasi-logarithmic scales are used for defining classes, as for the $K$ index (Mayaud, 1980). In Burlaga and Lazarus (2000), distributions of solar wind are fit to lognormal or double-peaked lognormal, since two solar wind dis-

tributions are employed, for slow and fast solar wind. Pulkkinen et al. (2012) use shifted lognormal for the geoelectric field amplitudes for past events. Love et al. (2016) employ 1-min horizontal component data from several observatories, fitting to threshold-truncated log-normal with least-squares and maximum likelihood method. Also Love et al. (2015) test both (truncated) lognormal and powerlaw *Dst* distributions, finding that different statistical tests favour the lognormal. Log-normal and powerlaw distributions are related, since some small changes in generative processes which show log-normal distributions may

lead to power-law generative processes (Mitzenmacher, 2003).

Power-law distributions are extensively used in many fields, mainly because they are scale invariant, e.g. in Newman (2005). A number of power laws appears in X-ray flare energy distribution, solar energetic particle event fluence distribution, auroral activity and radiation belts, due to the consideration of these physical systems as self-critically organized (e.g., Crosby, 2011). For solar flares, a number of power laws are obtained for flare length, volume, time and diffusivity, revealing the fractal

nature and the intrinsic relationship to self-organized criticality (SOC) (Aschwanden et al., 2013). In Yermolaev et al. (2013) the relationship between a variety of interplanetary features and *Dst* values is studied, fitting log-log quadratic functions, employing power law and square-law fits. Power laws are also utilized for relating the geomagnetic disturbance and its rate of change in scatterplots (Tóth et al., 2014).

Power laws appear as the most utilized distribution in Riley (2012) for CME speeds, X-ray flares and geomagnetic events.

Coincidentally, the *Dst* distribution index of -3.2 there is on the same order than the power law index for $|LCi\tilde{n}|$ distribution. The CME speed distribution also follows a power law with index equal -3.2. Kataoka (2013) fit power laws *dH* amplitude distribution, finding $\alpha$ around -3.2 (solar cycle 23) to -3.8 (solar cycle 16); this range is in agreement with the results obtained here. Wanliss and Weygand (2007) highlight the lack of characteristic time scale for power laws employed on storm waiting time with SYM-H index and other interplanetary medium parameters. They obtain an index slope of -1.2 using a powerlaw

with an exponential cutoff.

Power laws are also the most likely function for power spectra of a myriad of physical magnitudes: for solar wind characteristics, in Burlaga and Lazarus (2000) the power spectra of the solar wind is analysed, finding that the power spectral index for interplanetary medium density is $\alpha=-5/3$ (coincidental with that of Kolmogorov turbulence index, and also coincidental with the fractral powerlaw for 3D events and volumes (Aschwanden et al., 2013). For proton temperature $\alpha$ also gets values





in this range. The $\alpha$ parameter between –1 and –2 that they obtained may indicate small-scale structures in the solar wind fluctuations. In Zimbardo et al. (2008, and references therein) the power spectra index of the magnetic turbulence spectrum ($B^2$ vs. frequency) in the magnetosphere is investigated; more specifically, in the magnetosheath goes from –7/3 (implying turbulent cascade) to -2.6. Kiyani et al. (2015) analyse the different regimes in magnetic field fluctuations. In Burlaga and

Lazarus (2000) the spectrum for speed is around $\alpha$=–2.2. These $\alpha$ are on the same order than that obtained for the *LDiñ* power spectrum (plot not shown).

Next interesting distribution family is the Generalized Extreme Value distributions (GEV). The 'heavy-tailed distributions' as GEV distributions prove useful for taking into account the very extreme and unlikely events. The branch of GEV distributions (Fréchet, Gumbell or Weibull, depending whether the shape factor $c$ is larger than, equal to, or less than zero) is widely used

also (Thomson et al., 2011, see references therein). Here we have obtained some functions of this family, but not with low $D$, which means that they are not the best possible function.

In Tsubouchi and Omura (2007), the GEV distribution (a Gumbel distribution) is used in *Dst* for estimating the mean excess function, and the waiting time is estimated through a Poisson distribution (as made also by e.g. Kataoka (2013); Love (2012)). They also estimate an extreme value threshold of -280 nT by the different distribution regimes using a powerlaw index of -4.9

for the extreme-event tail. Nikitina et al. (2016) also employ GEV distribution for subauroral, auroral and polar geomagnetic data aiming at return period estimation. GEV distribution are also used for statistical distribution fit of flares in Curto et al. (2016). Thomson et al. (2011) use GEV in addition to a Generalized Pareto distribution.Wintoft et al. (2016) also apply GEV for the analyses of the return period on the derivative of the magnetic field and modelled electric field.

All these different works are dependent on data sample sizes, how evenly these data are distributed, and most importantly,

the fact that they are data from a variety of geomagnetic indices, sometimes as single-observatory sets, and sometimes as an average of different magnetometer data; therefore distributions applied may be different. The nature of the data values is also important: usually the geomagnetic components that are averaged on different observatories may exhibit smaller values than single-observatory data; the values would be even higher when single high-latitude observatories are considered.

*LDiñ* is an index from a single geomagnetic observatory, and the advantage of *LDiñ* is that the main regular variations and

trends have been effectively removed. This makes data, both the bulk and the tail, correspond to real geomagnetic disturbances, minimizing the amount of noise to be fit. It is also relevant to consider the length of the sequence: it consists of more than 8 million datapoints, comprising more than 15 years due to its 1-min cadence. To form an equivalent dataset *Dst* series of that size – or another 1-h index –, a 960-year series would be necessary, or 2880 years of 3-h $K_p$ or equivalent in cadence. The series employed here are considered a large number set in statistics.

For the $|neg(LDiñ)|$, both beta-prime (inverted beta or beta distribution of the second kind) and $f$- have the best fitting score. Mielke's distribution is also successful for fitting $|LCiñ|$ and exhibits a very good fit for $|neg(LDiñ)|$. This distribution is a beta-kappa kind. The best scores for $|neg(LDiñ)|$ are from the beta distribution family. The Pareto type-II distribution 'Lomax' scores poorly. This function is also related with Fisk (log-logistic) and Burr types. Some of the best fit presented here are beta functions, which are closely linked to the gamma function. Love (2012) uses discrete Poisson distributions that are

transformed into a gamma functions for frequentist and bayesian modelling.





For the $|LCi\tilde{n}|$, the powerlaw, Mielke (beta-kappa), Gilbrat (lognormal type), Fisk (log-logistic) best fits correspond to a variety of function families, noting that the fitting is in general worse than $|neg(LDi\tilde{n})|$ cases.

For the whole $LDi\tilde{n}$, the hyperbolic-sine Johnson-Su has similarity in shape compared to Student's $t-$ and its non-central version $nct-$, which is also a generalized hyperbolic distribution.

Both $f$- and beta-prime (inverted beta or beta distribution of the second kind) and power-law distributions are unbounded, in addition to the Johnson-Su (hyperbolic-sine type) and the Mielke (beta-kappa). This also applies to the power-law for $|LCi\tilde{n}|$, since it is not bounded either. This means that, statistically, the tails are not limited. Therefore, the best fit functions for $|neg(LDi\tilde{n})|$, $LDi\tilde{n}$ and $|LCi\tilde{n}|$ do not have upper limits, theoretically. Then, we may think in the variables plotted in the two-dimensional histogram shown in Fig 3: the $LDi\tilde{n}$ and $LCi\tilde{n}$ may not have any upper limit. This is in agreement with the

results in Wintoft et al. (2016). However, the physical parameters that trigger geomagnetic storms may be upper limited by the finiteness on impulsivity of solar features, such as CME speed, solar wind speed, shocks, which cannot take every kind of values, but have an empirical upper limit (finite size effect, e.g. Newman, 2005).

In addition to these considerations, thresholds are used in a reverse mode in this paper: first we fit data to a distribution and then we obtain thresholds. The usual reasoning, which is the utilization of ad-hoc thresholds for calculations, appear,

e.g., for mean residual life computation to estimate return periods, e.g., in Thomson et al. (2011), or as percentile comparison in Nikitina et al. (2016) in addition to the use of Q-Q plots, helping in the task of checking the goodness-of-fit of statistical distribution. Kataoka and Ngwira (2016) set some alert thresholds for geomagnetic disturbance and its rate of change, corresponding to different geomagnetic transients, without statistical distribution fit; or the percentile choice for proving the threshold independence of power laws in Wanliss and Weygand (2007).

As we mention before, no time between geomagnetic disturbances is considered. As future work, this can be considered for clustering proper geomagnetic storms; then, block-maxima related distributions to those obtained here would be expected.

## 5   Conclusions

Seeking the best distribution function to $|neg(LDi\tilde{n})|$ and $|LCi\tilde{n}|$ distributions, we found that the intersect points between the best fit function and the index CCDF can be used to establish thresholds, being more objective than some other ad-hoc

thresholds.

The best fits cluster in family distributions: for the $|neg(LDi\tilde{n})|$, beta distribution family is the predominant. In the case of $|LCi\tilde{n}|$, there is a variety of families, without a clear predominance. For the whole $LDi\tilde{n}$, hyperbolic distributions are the best fit functions. All these functions are unbounded.

The defined thresholds will appear in the Spanish Space Weather Service web-portal (SeNMEs, as in Palacios et al., 2016)

as a product in its final version, in http://www.senmes.es/index-en.php in quasi-real time along with a technique for translating real-time local geomagnetic indices into coloured scales that we created so users have an objective and timely depiction of recent geomagnetic disturbance: these thresholds are exhibited as the limits of the coloured scale marks in the background. Grey rectangles at the bottom of the colourbar indicate $LDi\tilde{n}$ and $LCi\tilde{n}$ values.



The main conclusions are listed as follows:

1. Contrary to other previous works, which state the lognormal as the usual and best statistical distribution for geomagnetic disturbances, we found that lognormal is not the best fit. The best fit distributions are those from the beta family for $|neg(LDi\tilde{n})|$ and hyperbolic family for the whole *LDiñ* distribution.

2. For the $|neg(LDi\tilde{n})|$, the best fits are for the betaprime, $f$-distribution, and Mielke (beta-kappa). The intersects with the first two functions yield crosspoints at –100, –205 and –475 nT, which can be utilized for establishing thresholds for intense, very intense and extreme disturbances. The different regime from distribution bulk and tail marked in –60 nT is considered for moderate disturbances. These thresholds can be useful for geomagnetic activity dependent users.

3. For the $|LCi\tilde{n}|$, the best fits are for the powerlaw, followed by Mielke (beta-kappa), Gilbrat (lognormal type) and Fisk (log-logistic) distributions. The powerlaw intersects the CCDF at 6 and 18 nT min$^{-1}$, determining the thresholds for unsettled and very intense disturbances in the geomagnetic field time derivative. For the extreme case, the intersection between the Gilbrat and betaprime yields a point in 32 nT min$^{-1}$. These thresholds can be useful for GIC-aware users.

*Competing interests.* We authors declare that they do not have competing interests.

*Acknowledgements.* The results presented in this paper rely on the data processed at the UAH and collected at observatory San Pablo de los Montes, Toledo, Spain. We thank IGN, for supporting its operation and INTERMAGNET for promoting high standards of magnetic observatory practice www.intermagnet.org.

We acknowledge the use of Python and their statistical packages for programming. ADS is also acknowledged for bibliography searching engine.

We thank funding from MINECO project AYA2016-80881-P (including AEI/FEDER funds, EU).



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





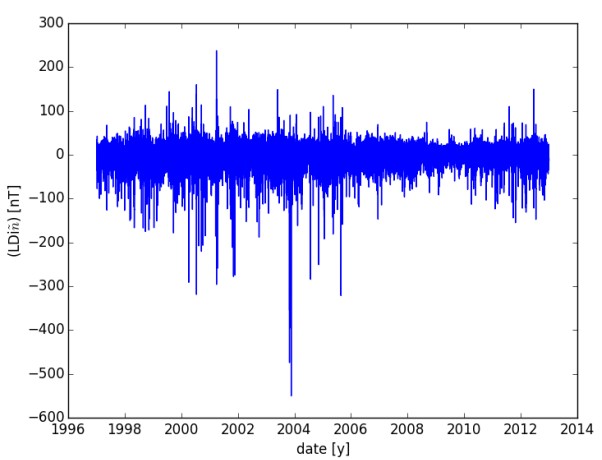 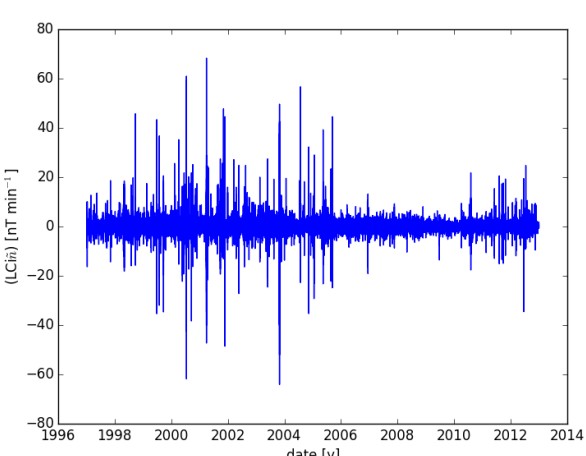

**Figure 1.** *Left*: Plot of *LDiñ* along time. *Right:* Plot of *LCiñ* along time.




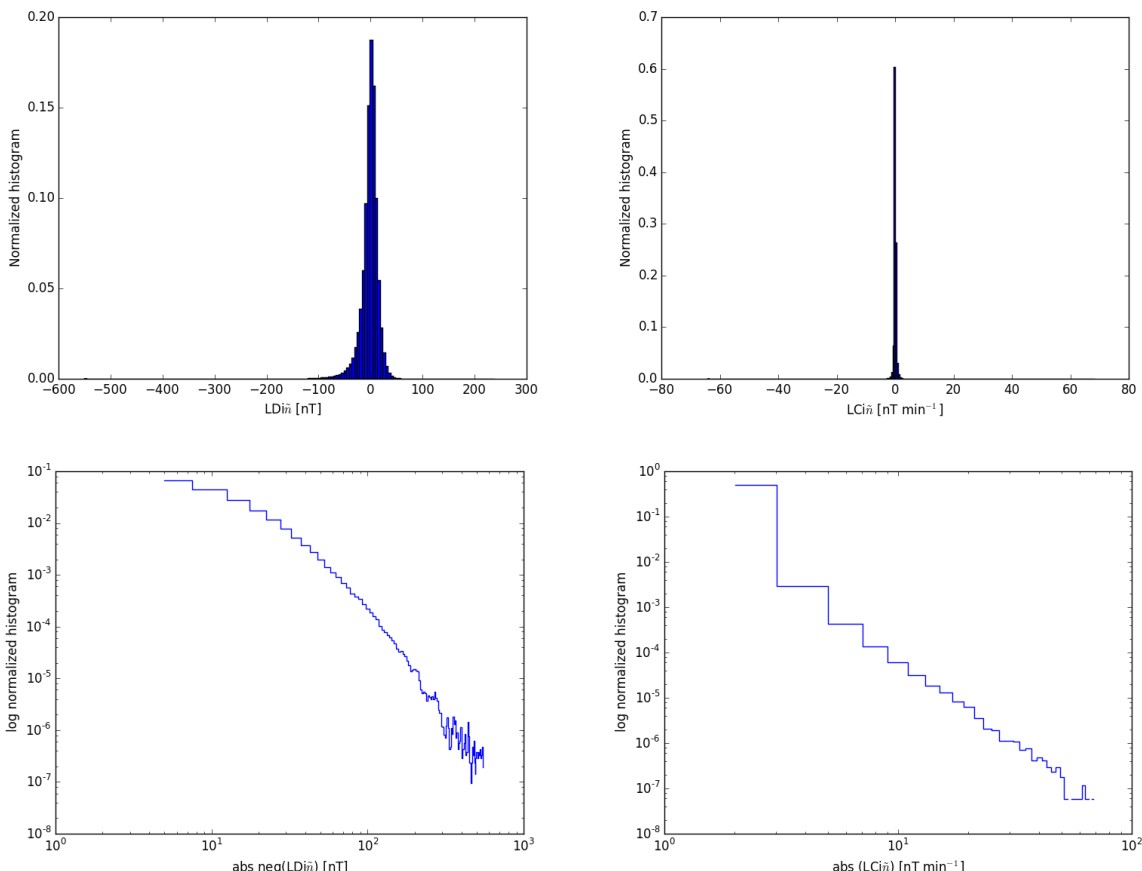

**Figure 2.** *Top left*: Normalized distribution of *LDiñ* values. Bin sizes of 5 nT. *Top right:* Normalized distribution of *LCiñ* values. Bin sizes of 0.5 nT min$^{-1}$. *Bottom left*: Log normalized distribution of $|neg(LDiñ)|$ values. Bin sizes of 5 nT. *Bottom right:* Log normalized distribution of $|LCiñ|$. Bin sizes of 2 nT min$^{-1}$.





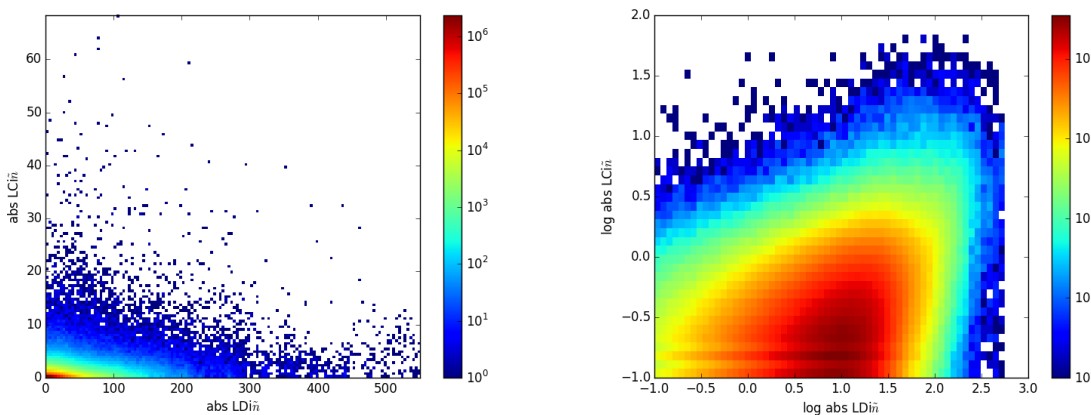

**Figure 3.** *Left:* two-dimensional histogram of the *LDiñ* vs. its derivative *LCiñ*. *Right*: log-log two-dimensional histogram – scatterplot of the
the *LDiñ* vs. its derivative *LCiñ*.





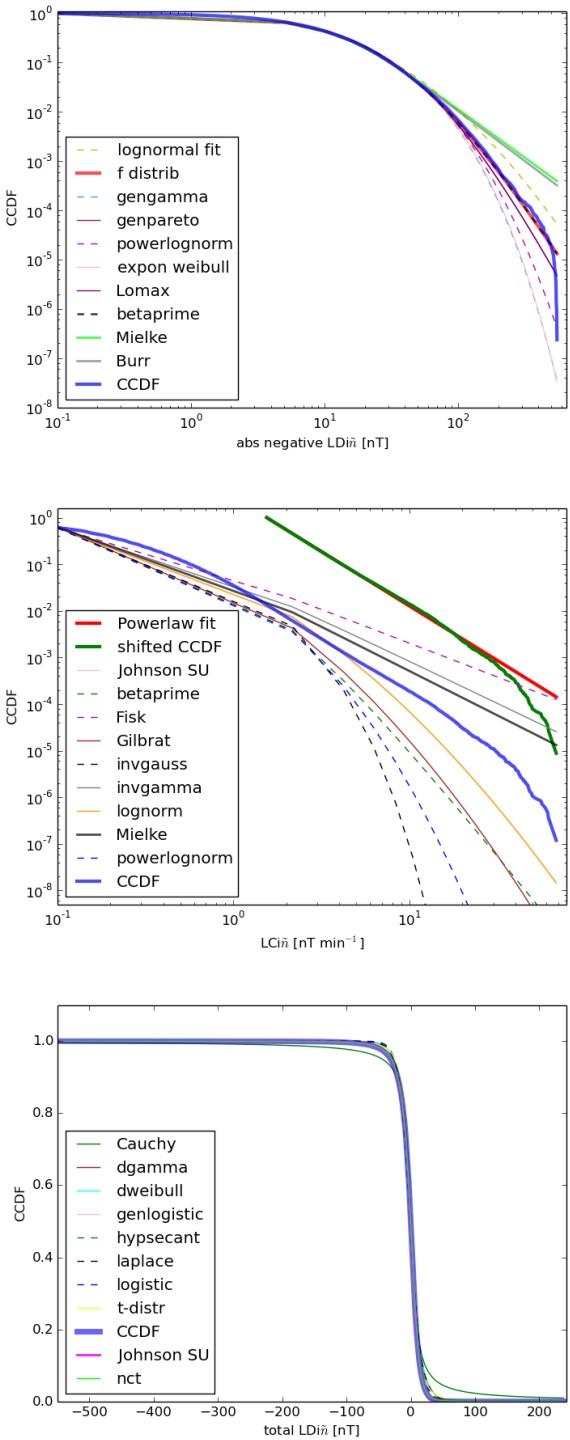

**Figure 4.** Best fit for different distributions. *Top:* Ten best fit distributions for $|neg(LDi\tilde{n})|$. *Middle:* Ten best fit distributions for $|LCi\tilde{n}|$. *Bottom:* Ten best fits for the whole *LDi\tilde{n}* distribution.