# Peer review of "Defining scale thresholds for geomagnetic storms through statistics"

_Natural Hazards and Earth System Sciences, 2017_

## Referee Comment (RC1) · M. A. Hapgood (Referee) · 29 Nov 2017

This manuscript analyses the statistical distribution of geomagnetic variability using a 15-year long high-quality and high-cadence dataset developed by the authors. They use this dataset to determine thresholds above which the Spanish Space Weather service should issue warnings that geomagnetic variability has reached high levels that may warrant actions by operators of systems (such as power grids) that are at risk from high geomagnetic variability.

This is an elegant approach, not least through the authors' use of a local geomagnetic index customised to their target service area in Spain, also through their study of a wide range of statistical functions to find a function that most suits the geomagnetic variabil-

ity observed in Spain. However, I consider that their approach, while mathematically consistent, is poorly suited to the practical task of managing the space weather risk to vulnerable systems:

a. most importantly there is no consideration of the system response to geomagnetic variability. This response is central to the setting of space weather thresholds. For example, a power grid operator will be concerned with the size and durations of geomagnetic induced currents passing through their transformers and how these are likely to degrade or even damage the transformers. This requires consideration of (a) how sub-surface geology (ground impedance) converts geomagnetic variability into geoelectric fields, and (b) how power grid topology determines the GIC flows driven by those geoelectric fields, especially if this leads to hot spots with high GIC (e.g. due to edge effects in the grid, and coastal enhancements of geoelectric fields). There are a number of published papers that have examined these issues in respect of the Spanish power grid, so I strongly encourage the authors to assess how their results can be linked to those studies and, in particular, to consider whether their risk thresholds need to be adjusted to levels of geomagnetic variability that can produce GICs that might challenge transformer and grid operations in Spain. I suspect that this is likely to require a substantial increase in the risk thresholds, but a final result will depend on comparison of the present work with existing literature on GIC flows in the Spanish power grid.

b. as a secondary issue, I note that the authors' risk thresholds are well below the 1-in-100 and 1-in-200 year levels customarily used by government risk managers and by insurance industry. If one considers that the target risk from space weather is akin the Carrington event of 1859, one should consider the likelihood of severe geomagnetic event over any particular location, such as Spain, with that event lasting up to one hour (as in the Mumbai/Colaba observations from 1859, also the failure of the Hydro-Québec grid in 1989). The 1-in-100 year likelihood of a single one-hour duration event is around one in a million, well above the threshold probabilities used by the authors. Thus I

recommend that they also consider adjusting their risk threshold to better match those widely used by emergency management authorities and the insurance industry – and to see how these thresholds compare with risk thresholds linked to system response as discussed under point a above.

For these reasons, I recommend that the authors make a major revision of the manuscript to bring it into better alignment with the wealth of published literature on space weather risks, especially to power grids including studies of space weather impacts on power grids in South Africa, Australia, Brasil, New Zealand, plus regions of the US and Europe. I include some examples below – but also see references in those papers. I would particularly highlight the South Africa and Australian studies as having magnetic latitudes, and perhaps geologies, that are comparable to Spain.

In addition I recommend that the authors address the minor comments and typographical issues listed below.

Minor comments

1. Page 1, line 24. This statement that geomagnetic disturbances decrease Earth's magnetic field is incomplete. This decrease is generally true where and when the disturbance is caused by the ring current, also by westward electrojets, but sudden impulses and eastward electrojets can produce increases that have very significant effects. Rotational disturbances of the field are also thought to have very important effects. Please update to reflect this.

2. Page 2, line 14. Suggest to cite the description of Dst, not as a plain URL, but rather as a reference to "Sugiura and Kamei, 1991" so as to recognise Sugiura's key role in developing Dst. Then reference as "Sugiura, M. and Kamei, T.: Equatorial Dst index 1957–1986, in IAGA Bull., 40, edited by A. Berthelier and M. Menvielle, ISGI Publ. Off., Saint-Maur-des-Fosses, France, 1991." I suggest to also note that this is available via the Kyoto URL than you have here.

3. Page 2, line 26. Please make clear than this raw K is calculated for a specific magnetic observatory, e.g. as shown at http://www.obsebre.es/en/currentvariationshortasj for daily K indices from the Ebro geomagnetic observatory.

4. Page 2, line 27. This is potentially confusing as written. The key point is that A is derived from a linearised version of K, specifically that 3-hourly K indices are converted to 3-hourly linear indices, denoted by lower case a - and only then averaged to derive a daily A index. Please clarify the text.

5. Page 2, line 29. The description of Ap is incomplete and perhaps confusing. The Ap index is the equivalent of Kp but follows a linear rather than a logarithmic scale. Both Kp and Ap exist in 3 hour and 24 hour versions. The 3 hour Ap is usually denoted in lower case, i.e. ap. Please clarify the text.

6. Page 2, line 35. I suggest to explicitly state that G1 to G5 is equivalent to Kp 5 to 9.

7. Page 3, line 6/7. Since you raise here the issue of high cadence, it would be appropriate here to introduce SYM-H as higher cadence development building on Dst. SYM-H is used later (page 9, line 2) without any introduction.

8. Page 3, lines 11/13. Please also cite work of other groups who have noted the importance of local disturbances, e.g, papers by Antti Pulkkinen and Chigo Ngwira (see list of reference below)

9. Page 10, lines 27 to 29. The discussion about dataset size may be mathematically correct, but does it have any significance for the physics or risk management? In risk management, e.g. assessment of flood risks, one typically requires a dataset that is at least 5 times longer than the longest return time that you wish to consider. This is a huge challenge for all space weather studies because our datasets are short, typically a few decades. Whilst the high cadence gives a lot of data, the dataset includes only a small number of significant space weather events, particularly the October 2003 events, and none of the larger historical events such as the Carrington event of 1859,

the Railroad Storm of 1921 or the March 1989 storm or some of the huge events in the 1940s and late 1950s. From a risk perspective, what matters is that the dataset includes a significant number of severe space weather events. In the present case, I think the discussion about dataset size is unhelpful and I recommend it be removed.

10. Page 11, line 15. I feel this neglects the importance of return period assessment. Return periods enable comparison with real world applications, so an important insight that can help designers of vulnerable systems such as power grids. For example, if there is a requirement for a transformer to have a design life of 50 years, it will help designers if we can estimate the probability of it being exposed to geomagnetic variability above some level during those 50 years. Please consider how to link your work to the assessment of return periods for severe events.

11. Page 11, line 16. Please explain briefly what are Q-Q plots, so that the reader does not need to search for an explanation.

Typographical issues

* Page 1, line 22, "do generate" may be better as "generates"

* Page 2, line 24, "originates" may be better as "leading to"

* Page 4, line 20, "compresses" may be better as "includes"

Some other power grid studies

Blake, S. P., P. T. Gallagher, J. McCauley, A. G. Jones, C. Hogg, J. Campanya, C. Beggan, A. W. P. Thomson, G. S. Kelly, and D. Bell (2016), Geomagnetically induced currents in the Irish power network during geomagnetic storms, Space Weather, 14, 1136–1154, doi:10.1002/2016SW001534.

Cannon, P., et al., 2013. Extreme space weather: impacts on engineered systems and infrastructure. Royal Academy of Engineering. http://www.raeng.org.uk/publications/reports/space-weather-full-report

Divett, T., Ingham, M., Beggan, C.D., Richardson, G.S., Rodger, C.J., Thomson, A.W.P. and Dalzell, M., 2017. Modeling Geoelectric Fields and Geomagnetically Induced Currents Around New Zealand to Explore GIC in the South Island's Electrical Transmission Network. Space Weather, 15(10), pp.1396-1412.

Hapgood, M. et al, 2016. Summary of space weather worst-case environments. Revised edition. RAL Technical Report RAL-TR-2016-006. https://epubs.stfc.ac.uk/work/25015281

Kelly, G. S., A. Viljanen, C. D. Beggan, and A. W. P. Thomson (2017), Understanding GIC in the UK and French high-voltage transmission systems during severe magnetic storms, Space Weather, 15, 99–114, doi:10.1002/2016SW001469.

Marshall, R.A., Kelly, A., Van Der Walt, T., Honecker, A., Ong, C., Mikkelsen, D., Spierings, A., Ivanovich, G. and Yoshikawa, A., 2017. Modeling geomagnetic induced currents in Australian power networks. Space Weather, 15(7), pp.895-916.

Matandirotya, E., Cilliers, P.J. and Van Zyl, R.R., 2015. Modeling geomagnetically induced currents in the South African power transmission network using the finite element method. Space Weather, 13(3), pp.185-195.

Matandirotya, E., Cilliers, P., Van Zyl, R.R., Oyedokun, D.T. and Villiers, J., 2016. Differential magnetometer method applied to measurement of geomagnetically induced currents in Southern African power networks. Space Weather, 14(3), pp.221-232.

Ngwira, C.M., Pulkkinen, A.A., Bernabeu, E., Eichner, J., Viljanen, A. and Crowley, G., 2015. Characteristics of extreme geoelectric fields and their possible causes: Localized peak enhancements. Geophysical Research Letters, 42(17), pp.6916-6921.

Pulkkinen, A., Bernabeu, E., Eichner, J., Viljanen, A. and Ngwira, C., 2015. Regional-scale high-latitude extreme geoelectric fields pertaining to geomagnetically induced currents. Earth, Planets and Space, 67(1), pp.1-8.

Pulkkinen, A., Bernabeu, E., Thomson, A., Viljanen, A., Pirjola, R., Boteler, D., Eichner,

J., Cilliers, P.J., Welling, D., Savani, N.P. and Weigel, R.S., 2017. Geomagnetically induced currents: Science, engineering, and applications readiness. Space Weather, 15(7), pp.828-856.

Torta, J.M., Marsal, S. and Quintana, M., 2014. Assessing the hazard from geomagnetically induced currents to the entire high-voltage power network in Spain. Earth, Planets and Space, 66(1), p.87.

Torta, J. M., A. Marcuello, J. Campanyà, S. Marsal, P. Queralt, and J. Ledo (2017), Improving the modeling of geomagnetically induced currents in Spain, Space Weather, 15, 691–703, doi:10.1002/2017SW001628.

---

## Referee Comment (RC2) · Anonymous Referee #2 · 19 Dec 2017

The authors propose a geomagnetic index that might be useful for monitoring geomagnetic activity that is hazardous to Spanish power-grid systems.

My primary concern for this manuscript is the complete lack of clarity of presentation something that is possibly related to the fact that the index of interest is patented (possibly for personal profit?).

I see that the LDiñ index discussed in this manuscript is cited to a Spanish patent (Guerrero et al., 2016). Is the nature of this index "proprietary" or is its nature open for other scientists to scrutinize? If it is proprietary, then I would suggest that its use and discussion in manuscripts to be published in a scientific journal is not appropriate. Note that the traditional scientific method, and one of the primary purposes of publication in a scientific journal, is open scrutiny by other scientists. This scrutiny helps to ensure

that scientific work contributes to positive forward progress.

I found the patent by Guerrero et al. on "google patents", where curiously there are no mathematical formulas describing the index. This concerns me. Normally geomagnetic indices are defined as clearly as possible, and this normally involves mathematical formulae.

At the very least, for a manuscript to be accepted for publication, the material should be presented with sufficient detail and clarity to permit reproduction by other investigators. And, yet, when we look at this manuscript, it is very difficult to understand what this index actually is. Since the index is patented, are other investigators prohibited from reproducing the index? If so, again, this would be contrary to the normal scientific method. So, again, I am concerned about whether or not this material is appropriate for a scientific journal.

With respect to more minor points ...

Abstract should be a terse summary of results. It should not contain introductory material.

Page 1, Line 3. Indices don't inherently "have scale thresholds to quantify the severity or risk", but, rather, we humans might seek to make such assignments to the data. Page 1, Line 6, For which latitude are these indices applicable? Page 1, Line 10, what is "beta prime"? Page 2, Line 5, a reference for summarizing deleterious effects is needed. Here is a suggestion:

@book{13canetal, author = "P. Cannon and M. Angling and L. Barclay and C. Curry and C. Dyer and R. Edwards and G. Greene and M. Hapgood and R. B. Horne and D. Jackson and C. N. Mitchell and J. Owen and A. Richards and C. Rodgers and K. Ryden and S. Saunders and M. Sweeting and R. Tanner and A. Thomson and C. Underwood", title = "{Extreme Space Weather: Impacts on Engineered Systems and Infrastructure}", year = "2013", publisher = "Roy. Acad. Engineer.", address = "London, UK", pages =

"1–68" }

Page 3, Line 15, the word "spike" often denotes an artificial signal. One might instead use the word "impulse" or "rapid variation". Page 5, Line 20, why are statistical distributions being named with such unconventional labels?

---

## Author Comment (AC1) · 4 Jan 2018

Dear M.A. Hapgood (Reviewer 1)

Thank you for your careful revision of the manuscript. Your suggestions will improve its quality. All remarks are carefully explained below.

**General comments**

a. Thanks for the comment. We have included the following explanation to link with the following paragraph and explanation to link the GIC production, as an effect of the adverse space weather conditions, with the cause, that are the geomagnetic disturbances. It also appears summarized in the corrected version of the

manuscript.

The derivative of *LDiñ*, named *LCiñ*, is highly correlated with the geomagnetically induced currents records from the Spanish Power Company (Cid et al., 2016). Indeed, the recorded GICs, which depends on the substation, show a linear relationship with the *LCiñ*. This links GICs with the index derivative.

Being aware of these facts, and knowing that accurate real-time monitoring according to the needs of the final users is key, the Spanish Space Weather Service (SeNMEs) in 2014 introduced the G- and C-scale, for *LDiñ* and *LCiñ*, respectively.

The G- and C- scales in SeNMEs are related to the natural phenomena involved, and not directly related to the potential consequences (it is not a effect-based scale). To properly suit to the practical task of managing the space weather risk to vulnerable systems, every system shall establish their own risk protocols based on the potential consequences expected when a threshold is surpassed.

As an example, the differences between a scale related to a natural phenomenon and that related to the risk can be understood considering weather and climate events, for example, extreme rainfall. The occurrence of a value of total cumulative precipitation at a given time scale above a threshold value near the upper end of the range of observed values will be considered as an extreme rainfall event. However, the potential occurrence of flooding, including risk to human life, damage to buildings and infrastructure, will not only depend on the total cumulative precipitation, but also, on the orography of the place where the rain is falling.

b. Thanks for the comment. We have revised the literature provided by Reviewer 1. For comparison between the GICs measured in Spain and others measured in equivalent magnetic latitudes, we refer the reader to Cid et al. 2016, STO-MP-SCI-283 (already referenced in the manuscript), which shows very similar GIC values to Matandirotya et al. 2016, minding those 5-min averages on GIC values. We have updated the text to include the reference of Ngwira et al. 2015,

which quotes the papers by Pulkkinen et al. 2015, and other authors who have noted the importance of local disturbances. Some of these references does not seem to be directly comparable, as Divett et al. 2017, where New Zealand in the southern hemisphere is at the same geomagnetic latitude as Edinburgh and it is a slender island; or Marshall et al. 2017, about GICs in Australia (this country is very large, with important differences in latitude). Some other references on modelled GICs may yield overestimated values, and direct GIC measurements are more scarce. Most importantly, we may consider that the power network structure and its geological and geographical differences are some of the most relevant factors on GIC production.

About the thresholds mentioned in the literature, we have to remind that usually they are defined as return periods on 100- 1000-y, or used to compute the return periods through a threshold, defined ad-hoc high enough to be used by the peak-over-threshold method, and then fitted with a Generalized extreme value distribution, GEV (Pulkinnen et al. 2015). Most of the Reviewer 1's recommended references are thresholds derived for high latitudes, or different industry safety levels and ranges. Even the same geomagnetic latitude cannot guarantee an equivalence in the GIC magnitude order, since they are very dependent on power grid and geology.

As Reviewer 1 has noticed, the method described in the manuscript is applicable to any data range, amount, and index, since only data-related best fit distributions and their intersects will define the thresholds.

Please also refer to Points 9 and 10.

**Specific comments**

1.- Thanks for the comment. The text has been updated to reflect it.

2.- Thanks for the suggestion of Sugiura and Kamei, 1991. It has been properly included.

3.- Thanks. It has been changed accordingly to emphasize that is an index that can be computed for any observatory.

4.-5. Thanks for the comment. These two points have been clarified in the text. The corresponding comments on $K$ and $A$, $A_p$ and $a_p$ have been included in the corresponding paragraph.

6.- $G$ scale and $K_p$ equivalence is set in the text in the Introduction.

7.- Now $SYM - H$ has been presented in the Introduction. To make the formulation of *LDiñ* clearer, we have included a paragraph in the 'Geomagnetic data' section comparing *LDiñ* to $SYM - H$ in selected storms.

8.- Thanks. We have added some of the suggested literature to the manuscript (see (b)).

9.- Thanks for the comment. Unfortunately we do not agree on the comment of removing the explanation about datasets. Statistically it is very important the data number and the subsequent distribution shape, and it can be modified due to the addition of more data. We can explain with a example: a sample of photons that arrives to a detector, when the amount is scarce, can be fitted by a poissonian (it is the assumed typical shape). However, when an important amount of photons gets into a detector, the distribution will take probably a gaussian shape. Therefore the amount of course modifies the distribution shape, and subsequently, many other parameters such the return period, which is directly dependent on the distribution choice.

We agree on the wealth of literature describing GICs and the recent advances on statistics that are directly applied to this field. Again, different distribution shapes may arise depending on the magnetic latitude, therefore thresholds should be local to be more precise. Reviewer 1 refers to some specific statistical treatments: when a binomial or peak-over-threshold is estimated, sometimes with clustering

involved, only a limited number of events arise and it complicates the statistical treatment, as samples are more limited that way. It is not the case for the method presented in the manuscript.

10.- Considering the thumb rule of 5 times the data period, we could not reliably compute a 100-y return period since we have 15 years of data. Return periods are usually defined on binomial but recently poisson distribution or generalized extreme values distributions (considering peak over thresholds) have been used. It is computed directly by cumulative distribution functions CDFs through SFs (survival functions). These return periods can be considered as extrapolations of the distributions, and some caution is required when any scenario is projected with them. These scenarios may be better suited when fed with long-term solar and geomagnetic field data. This is the reason why we appreciate them but we do not focus on return periods.

11.- Q-Q plots have been introduced in the Discussion.

**Technical corrections**

Typographical issues.

1. Corrected.

2. We think it is meant on page 2, line 4. Corrected anyway.

3. Corrected. We meant 'comprises'.

Best regards,

The Authors

Please also note the supplement to this comment:

https://www.nat-hazards-earth-syst-sci-discuss.net/nhess-2017-367/nhess-2017-367-AC1-supplement.pdf

---

## Author Comment (AC2) · 4 Jan 2018

**General comments**

Dear Anonymous Referee 2,

About the patent topic, we should make a number of points clear:

First of all, patents are publicly available after the evaluation period, as it is our case. Probably the referee may need an explanation on how a patent procedure works. When a patent is sent for evaluation, it is checked on worldwide web databases and also in publications and technical notes about the topic. Patent is checked for clarity and concision (therefore all necessary math is explained in the patent text), with a number of iterations that has improved its quality. Therefore, the patent process is actually more

strict than any other publication, and it has to guarantee novelty. After the evaluation period, patents can be checked and its content may be reproduced, but under certain conditions. A patent is other way to protect intellectual property, as any other applicable to a publication.

About the work clarity, the paper includes all the usual sections for a scientific article, and they are clearly explained and referenced. To improve the clarity, a new part comparing the new index with others has been added.

Second, the own patent's authors, by definition, cannot get any profit using data from their own patent.

Third, it is not stated as any kind of Ethical Issue at NHESS or its Editorial, neither in competing interests, to use an own patent for producing own data. The patent is properly referenced and mentioned on the paper.

The fourth point is that data produced under a patented method can be available under request. Embargoed data exist in every field but it is always assumed as admissible to produce scientific results and publications with them. In addition to this, different kind of proprietary data are used for scientific purposes. For instance, Earth observation data are usually obtained by purchase, and subsequently used for publication, as multispectral ground observation or meteorological data.

**Specific comments**

Fifth, and most importantly, it is clearly stated (several times on the paper) that the method presented on this work can be applied to any other index; so, whoever is interested, may apply the method. The purpose of this paper is not 'proposing a new index', something obvious from the abstract to the end. All the previous discussion actually obscures the goal of the paper by a good amount.

**Technical corrections**

Minor points:

- About the Abstract, we consider that it is appropriate to put an introductory sentence about the natural hazard involved, since it is a multidisciplinary Journal.

- Page 1, Line 3: "These indices have some scale thresholds" has been substituted by "These indices usually have some associated scale thresholds"

- Page 1, Line 6: regional is mid-latitude. This has been incorporated to the Abstract.

- About the deleterious space weather effects, some references are already mentioned at that paragraph.

- Page 3, Line 15: For a 'spike' explanation (since they are not artificial spikes but 'H-spikes'), please refer to Cid et al., 2015; Saiz et al., 2016, as already mentioned in the text.

About the distribution nomenclature (Page 1, Line 10; Page 5, Line 20), they are well described in the Discussion. Therefore please refer to https://docs.scipy.org/doc/scipy-0.18.1/reference/stats.html mentioned in the text. Anyway they have been clarified in the Abstract.

If you have any further suggestion that can actually improve the paper, please let us know. Thanks for your time.

Best regards,

The Authors

Please also note the supplement to this comment:
https://www.nat-hazards-earth-syst-sci-discuss.net/nhess-2017-367/nhess-2017-367-AC2-supplement.pdf

[Figure]

**Supplement:**

**Defining scale thresholds for geomagnetic storms through statistics**

[revised manuscript text omitted]

Furthermore, the derivative of *LDiñ*, named *LCiñ*, is highly correlated with the geomagnetically induced currents records from the Spanish Power Company (Cid et al., 2016). Indeed, those recorded GICs, which depends on the substation, show

a linear relationship with *LCiñ*. Being aware of these facts, and knowing that accurate real-time monitoring according to the needs of the final users is key, the Spanish Space Weather Service (SeNMEs, http://www.senmes.es/index-en.php) in 2014 introduced the G- and C-scale, for *LDiñ* and *LCiñ*, respectively. The G and C scales in SeNMEs are physical-based scales, not effect based ones. To properly suit to the practical task of managing the space weather risk to vulnerable systems, every system

5    shall establish their own risk protocols based on the potential consequences expected when a threshold is surpassed.

Thresholds are not only employed to classify the severity of a natural hazard but also employed for computing the waiting time and time occurrence of events. The thresholds used for several types of natural hazards such as earthquakes, volcanic eruptions or floods are based on 1 in 50- or 1 in 100-year extremes; this approach for space weather is also applicable, as shown in different works, e.g., Love (2012); Nikitina et al. (2016); Love et al. (2016); Wintoft et al. (2016), among others.

10    In this paper, we use different distribution functions seeking the best fit for the regional index *LDiñ* and its derivative *LCiñ*; then, we define thresholds for this index and its derivative. Fitting distributions aiming at recurrence or waiting times estimations is out of the scope of this work. Opposite to the vast majority of the mentioned references, we do not use thresholds to help fitting a part or the whole distribution; threshold rationale and distribution properties are the main purposes. Hence, one of the key goals in this paper is aiming at overcoming some of the arbitrariness to establish thresholds for geomagnetic

15    disturbances and its rate of change through the objective method explained in this paper. Its application can be extended to any other global or regional index.

The paper is structured as follows: data used are presented in Section §2, along with the preliminary statistical considerations; next in Section §3, the method is described, obtaining the best fit distributions to the indices in Section §3.1; and then in Section §3.2, the thresholds for these indices are defined by these statistical best distribution intersects. The results and

20    discussion are presented in Section §4, and Conclusions appear in Section §5.

**2   Geomagnetic data**

We used *LDiñ* data computed from SPT geomagnetic observatory (San Pablo de los Montes, Toledo, Spain) from 1997 to 2012 – solar cycles 23 and 24, comprising two rising phases and one solar maximum –, processed in such a way to remove the daily variation in the $H$ component of the geomagnetic field, which is not straightforward for mid-latitude data. The processing

25    method, which is now patent pending (Guerrero et al., 2016), will be actually detailed in a forthcoming paper.

Therefore we consider *LDiñ* as this local geomagnetic index after this specific processing, whose units are nT and its temporal cadence is 1 minute. We define hereafter *LCiñ* as the temporal derivative of *LDiñ*, in nT min$^{-1}$. Since the goal of this paper is not the storm recurrence or waiting time, not timing between peak values is considered. The whole dataset are 1-min values without any further consideration of recurrence.

30    To prove the validity of *LDiñ* for monitoring local geomagnetic disturbances while effectively removing other trends, we show two examples of geomagnetic storms with 1-min cadence indices in Fig. 1. In the left panel, *LDiñ* (blue), the unprocessed data from SPT (green) and *SYM-H* (red) from December 12-16, 2015. Quiet days prior to the storm have the solar regular daily geomagnetic variation corrected in *LDiñ*, and all phases of the storm are kept with this index, which do not occur on the other

records. In the right panel, we show data from *LDiñ*, unprocessed data from SPT and *SYM-H*, from October 29 to November 1, 2003. The H-spike is clearly shown in the local records *LDiñ* and unprocessed SPT. The storm main and recovery phase are well conserved. The *SYM-H* index does not exhibit the H-spike, it is clearly missed due to the average in longitude; and even main phases are different because of the data combination from different geomagnetic observatories. This fact is also observed in the 2015 storm.

[Figure]

**Figure 1.** *Left*: December 2015 storm recorded by *LDiñ*, unprocessed data from SPT and *SYM-H*. *Right:* October 2003 storm *LDiñ*, unprocessed data from SPT and *SYM-H*.

We computed more than 15 years of data. This period is chosen since it is almost the whole period in which SPT has their data digitized. Also, it  comprises the solar cycle 23 and its very intense geomagnetic storms. The *LDiñ* and its temporal derivative along almost 16 years are plotted in Fig. 2. The values are upper limited (in absolute value) to the highest peak values of the geomagnetic disturbances during the solar cycle 23. Values of the derivative are also constrained by these storms.

To give a hint about the value distributions of *LDiñ* and *LCiñ*, we plotted the histograms, normalized to the total amount, in Fig. 3. The bin size is 5 nT for *LDiñ* and 0.5 nT min$^{-1}$ bin size for *LCiñ*. On geomagnetic storms, negative values actually define the storm since they mean a horizontal component depression of the geomagnetic field due to a ring current enhancement; therefore we will take into account only these negative values, naming it as $|neg(LDiñ)|$. It is evident that this distribution is not gaussian but other kind of distribution, with a long tail and very skewed to negative values, as shown in in Fig. 3 (top left).

We consider that the positive and negative values of *LCiñ* are meaningful because both may be related to the geomagnetically induced current (GIC) generation, interesting for space weather purposes. Therefore, we will consider this possibility equally for both signs by studying the absolute value of *LCiñ*, naming it $|LCiñ|$. As shown in Fig. 3 (top right), the distribution of *LCiñ* has a very small skewness.

Taking into account the significant values in the form of $|neg(LD\tilde{n})|$ and $|LC\tilde{n}|$, we plotted *log-log* histograms. In the case of $|neg(LD\tilde{n})|$, the heavy tail of the distribution is shown clearly in the bottom left of Fig. 3. A conspicuous linear trend appears for $|LC\tilde{n}|$ histogram in the bottom right panel of Fig. 3. We note the bin size variation in logarithmic scale, issue that will be improved in the next Section.

5  We also show a scatterplot of the absolute value of *LD\tilde{n}* and *LC\tilde{n}* to see how their values relate to each other and show the relationship, if any, between extreme values. Unfortunately we were not able to fit a line of this scatterplot due to the large amount of points, similar to those in Tóth et al. (2014). Due to the large dataset, we created a two-dimensional histogram instead, plotting the absolute values of *LD\tilde{n}* and *LC\tilde{n}*, in the left panel of Fig. 4. The bin size is equivalent to $\approx 6$ nT for *LD\tilde{n}* and 0.5 nT min$^{-1}$ for *LC\tilde{n}*. The histogram evidences that large values of $|LC\
[revised manuscript text omitted]